# Chicken Meal as a Fishmeal Substitute: Effects on Growth, Antioxidants, and Digestive Enzymes in *Lithobates catesbeianus*

**DOI:** 10.3390/ani14152200

**Published:** 2024-07-29

**Authors:** Bo Zhu, Wenjie Xu, Zhenyan Dai, Chuang Shao, Yi Hu, Kaijian Chen

**Affiliations:** 1Fisheries College, Hunan Agricultural University, Changsha 410128, China; bp.zhu@outlook.com (B.Z.); xwjxqsz@163.com (W.X.); dzy7486@163.com (Z.D.); 13605186516@163.com (C.S.); 2Hunan Engineering Technology Research Center of Featured Aquatic Resources Utilization, Hunan Agricultural University, Changsha 410128, China

**Keywords:** fishmeal, growth performance, antioxidant, intestinal inflammation, *bullfrog*

## Abstract

**Simple Summary:**

Fishmeal is a vital protein in aquafeed; however, its high cost and limited availability pose significant challenges. This study investigates the efficacy of chicken meal, an animal protein with a higher yield and lower cost, as a substitute for fishmeal. After feeding bullfrogs with different feed formulations for eight weeks, it was found that replacing 50% of fishmeal with chicken meal increased the survival rate of bullfrogs, thus enhancing their overall weight. Additionally, this partial substitution improved the antioxidant capacity and intestinal digestive enzyme activity of the bullfrogs. Conversely, a 100% replacement of fishmeal showed no significant effect on survival rate and resulted in reduced weight gain and lower amino acid content in muscle tissues, indicating that complete replacement is unsuitable.

**Abstract:**

In pursuit of sustainable aquaculture, this study was performed to evaluate chicken meal as a substitute for fishmeal in bullfrog diets. Three experimental groups were established: a control group (FM) with 20% fishmeal, a CM50 group with 50% replacement (10% fishmeal), and a CM100 group with 100% replacement (0 fishmeal). Bullfrogs were fed for 56 days. The CM50 group exhibited significant increases in total weight gain and survival rate and a notable decrease in feed coefficient (*p* < 0.05). However, the CM100 group showed contrary effects. Increasing chicken meal substitution correlated with decreased amino acid content in muscle. Notably, the CM50 group demonstrated enhanced activities of antioxidant enzymes (CAT, T-AOC) and elevated gene expression levels (*cat*, *sod*, *gst*, etc.) in muscle and the intestine (*p* < 0.05), improved intestinal morphology, enhanced digestive enzyme activities (amylase, lipase), and reduced expression of inflammatory factors (*il-1β*, *il-8*, *il-17*, etc.). Conversely, the CM100 group’s indicators regressed to levels similar to or worse than those of the FM group. Therefore, a 50% substitution of fishmeal with chicken meal effectively promoted bullfrog survival, protected the intestines, and enhanced antioxidant capacity, supporting its potential as a fishmeal alternative. However, the adverse outcomes of the CM100 strategy, including growth retardation and reduced amino acid content in muscle, indicate that complete replacement is unsuitable.

## 1. Introduction

The rise of aquaculture has effectively alleviated the pressure on human food resources [1,2]. As a leader in this field, China’s aquatic product exceeded 71 million tons in 2023, with an annual growth rate of 3.4%. Of this, the output from artificial breeding was 58 million tons [Ministry of Agriculture and Rural Affairs of the People’s Republic of China, 2023]. Artificially cultivated fish typically rely on feed, which has led to a surge in demand for high-quality feed protein, especially fishmeal [3,4]. However, given the depletion of marine resources and rising costs, exploring sustainable alternatives to fishmeal is imperative [5,6]. In recent years, researchers have actively explored a variety of new proteins, such as plant protein, insect protein, and single-cell protein, as potential substitutes for traditional fishmeal, demonstrating considerable potential in ensuring the growth performance of aquatic organisms [7,8]. Despite these advancements, the application of these proteins still faces challenges, including the inherent anti-nutritional components and unbalanced amino acid composition of plant-based proteins, which may affect the digestion, absorption, and overall health of animals [9,10,11]. Additionally, while single-cell proteins are highly nutritious, their high costs limit their widespread use in commercial feeds [5,12]. Therefore, although research on alternative proteins continues to progress, the market has yet to see an ideal substitute that can fully match the performance of fishmeal, highlighting the need for continued exploration and optimization.

Chicken meal, as a high-value protein from animals, has become a research focus in academia due to its nutritional value and balanced amino acid composition [13,14]. Extensive research evidence indicates that it can effectively replace fishmeal in various aquatic species, particularly in well-designed amino acid-balanced formulas, showing the potential to maintain or even enhance the growth performance of aquatic animals [15,16]. Moderate substitution of fishmeal not only ensures unimpeded growth but has also been found to actively promote metabolic activities, optimize intestinal health, and reflect its additional value in improving the welfare of aquaculture animals [17,18,19]. Importantly, the application of chicken meal, as an upgraded utilization of chicken processing by-products, highlights the concept of a circular economy. This approach circumvents the ethical dilemma of competing directly with humans for food resources, while effectively recycling and transforming food industry waste. It significantly reduces waste within the food chain and alleviates the environmental burden. This strategy demonstrates an innovative practice in promoting food security and environmental protection.

Bullfrogs hold an important position in China’s aquaculture, with artificial breeding output exceeding 500,000 tons in 2021, showcasing the industry’s vigorous development [20]. However, research on the refinement of bullfrog feed formulas and fishmeal substitution strategies is still in its infancy, with limited scientific literature. Given the carnivorous nature of bullfrogs, seeking efficient and sustainable fishmeal substitute resources, such as chicken meal, is particularly urgent for maintaining and promoting the long-term development of the bullfrog farming industry. This study is based on this urgent need and systematically examines the actual efficacy of chicken meal as a substitute for fishmeal from multiple dimensions, including bullfrog growth performance, muscle quality, and intestinal health. This research aims to provide empirical evidence for the scientific optimization of bullfrog feed formulas, fill existing research gaps, and promote more robust growth in the bullfrog farming industry while respecting environmental carrying capacity.

## 2. Materials and Methods

### 2.1. Experimental Feed

Chicken meal (CM) is a high-protein ingredient derived from slaughter by-products, including chicken carcasses and internal organs. The production process involves high-temperature steaming for sterilization and defatting, followed by spray drying. The resultant product is then ground and sifted to achieve an appropriate particle size for incorporation into the feed of animals. For this study, the chicken meal, provided by New Hope Group, contained 69.39% crude protein and 13.37% crude lipid. This research, a collaborative effort between Hunan Agricultural University and New Hope Group, investigated the feasibility of substituting fishmeal with chicken meal in bullfrog feed. As of June 2024, the import price of chicken meal in China is approximately USD 1130 per ton, compared to around USD 2200 per ton for Peruvian super fishmeal, thereby making chicken meal a cost-effective alternative. The feed employed in this study has been adapted from a commercial formulation provided by New Hope Group, meeting practical requirements.

This experiment was designed with three feed formulations to investigate the impact of substituting chicken meal for fishmeal on the rearing of bullfrogs. The specific groupings were as follows: (1) control group (FM), containing 20% fishmeal; (2) partial replacement group (CM50), where fishmeal was halved to 10% and supplemented with a corresponding proportion of chicken meal; (3) full replacement group (CM100), where fishmeal was completely replaced with chicken meal, resulting in 0 fishmeal content. For detailed ingredient and amino acid compositions, refer to Table 1 and Table 2.

The experiment utilized 4 mm diameter extruded floating pellets as the form of feed. All raw materials were supplied by New Hope Group Co., Ltd. (Chengdu, China), and the experimental team collaborated with the company’s engineers to complete the meticulous formulation and processing of the feed at its headquarters in Chengdu, Sichuan.

### 2.2. Animals and Feeding Management

This feeding experiment was carried out in Changde City, Hunan Province, utilizing canvas pool cages designed to replicate the local environment. Juvenile bullfrogs were acquired from a nearby farm and acclimated at the experimental site for three days before this study commenced.

For the experiment, 1200 bullfrogs of uniform size were randomly allocated to 12 cages, each measuring 0.8 m × 0.8 m × 0.6 m, with 100 individuals per cage. These 12 cages were divided into three groups, with four cages per group, each receiving a distinct diet. Following a 24 h fasting period, the bullfrogs were fed twice daily at a rate of 3~5% of their body weight (at 7:00 and 17:00). The feeding quantity was adjusted every four days based on the daily temperature and the amount of leftover feed 10 min after feeding, ensuring the bullfrogs were adequately satiated. Every ten days, feeding was halted for one day, and the cages were cleaned using a siphon method to maintain environmental hygiene. Mortality and weight of the bullfrogs were recorded daily throughout the 56-day experiment.

Water quality parameters were meticulously maintained, with an average air temperature of 24.0 °C, water temperature of 21.8 °C, dissolved oxygen levels of at least 4.0 mg/L, pH values ranging from 7.6 to 7.8, ammonia nitrogen levels below 0.4 mg/L, and nitrite nitrogen levels below 0.005 mg/L. These conditions ensured the stability and scientific validity of the experimental environment.

### 2.3. Sample Collection

After the 56-day feeding trial, bullfrogs were subjected to a 24 h fasting period. Subsequently, precise counts and total weight measurements were taken for the bullfrogs in each cage. Each bullfrog was then anesthetized with MS-222 to minimize discomfort, following ethical standards. This study was approved by the Animal Care Committee of Hunan Agricultural University and adhered to the Chinese Animal Welfare Guidelines.

From each cage, four bullfrogs were randomly selected, and their leg muscles were dissected and frozen for analysis of amino acid content. An additional four bullfrogs per cage were selected for sampling approximately 1 cm of the mid-intestine. These samples were gently rinsed with a 0.7% saline solution and fixed in a 4% paraformaldehyde solution to prepare HE-stained intestinal tissue sections.

Additionally, six bullfrogs were randomly selected from each cage, and samples were collected from the left thigh muscle and mid-intestine, ensuring the latter was free of feces and debris. These samples were placed into 1.5 mL enzyme-free centrifuge tubes, rapidly frozen with liquid nitrogen, and stored for subsequent enzyme activity assessments and gene expression analysis.

### 2.4. Sample Analysis, Detection, and Calculation

Growth performance indicators were calculated as follows:(1)$$Final\;average\;weight\left(g\right)=\left(\frac{\;final\;total\;weight}{final\;count}\right)$$
(2)$$Average\;weight\;gain\;rate\left(\%\right)=100\times \;\left(\frac{\;final\;average\;weight-initial\;average\;weight}{initial\;average\;weight}\right)$$
(3)$$Total\;weight\;gain\;rate\left(\%\right)=100\times \left(\frac{\;final\;total\;weight-initial\;total\;weight}{initial\;total\;weight}\right)$$
(4)$$Feed\;coefficient=\left(\frac{\;total\;feed\;intake}{final\;total\;weight-initial\;total\;weight}\right)$$
(5)$$Survival\;rate\left(\%\right)=100\times \left(\frac{\;final\;count}{initial\;count}\right)$$

The crude protein content in the feed was determined by the Kjeldahl method. Initially, the samples were mixed with sulfuric acid and a catalyst and heated on a hot plate until complete digestion occurred, converting the nitrogen in the proteins into ammonium ions. Subsequently, the ammonium ions were distilled to form ammonia gas, which was absorbed by a boric acid solution. Finally, the absorbed solution was titrated with a standard acid solution, and the nitrogen content was calculated based on the volume of acid used, which allowed for the determination of crude protein content. The crude lipid was detected using a Soxhlet extractor. The samples were placed in an extraction thimble, and the lipid was repeatedly extracted using a cyclic process with an organic solvent (petroleum ether). After the extraction, the solvent was evaporated. The crude lipid content was calculated by measuring the weight difference of the samples before and after extraction.

The composition and content analyses of amino acids were performed using high-performance liquid chromatography (HPLC), which was also applied to the amino acid samples of the feed and bullfrog muscle. This method was applied to both the feed and bullfrog muscle amino acid samples. The samples were separated through a chromatographic column in the HPLC system. Due to the distinct physicochemical properties of various amino acids, they were retained in the column for different durations, allowing for their separation. The separated amino acids were detected by a detector, producing corresponding chromatographic peaks. By comparing these peaks with standard samples, the types and concentrations of amino acids in the samples were determined. The antioxidant performance indicators (including peroxidase, superoxide dismutase, total antioxidant capacity, and malondialdehyde) and intestinal digestive enzyme activities (amylase, lipase, trypsin) were measured using specialized reagent kits provided by Nanjing Jiancheng Bioengineering Institute Co., Ltd. (Nanjing, China).

Tissue samples were rinsed with running water for 6 h, followed by gradual dehydration with alcohol and clarification with xylene, and then embedded in paraffin, sectioned, and stained to prepare intestinal tissue sections. Finally, morphological observations and analyses were conducted using an optical microscope and the Slide Viewer image analysis system.

RNA was extracted from bullfrog muscle and intestinal tissues using the Trizol method. To assess RNA integrity, 1.0% agarose gel electrophoresis was used, while a nucleic acid analyzer measured RNA concentration. Qualified RNA samples were reverse-transcribed to synthesize cDNA, followed by qPCR analysis. Details of the primer sequences used for the qPCR experiments can be found in Table 3.

### 2.5. Statistical Analysis of Data

Data processing and statistical analysis were completed using SPSS Statistics 24 software. Initially, the collected experimental data underwent tests for normality and homogeneity of variance. Upon confirming that the data conformed to a normal distribution and homogeneity of variance, a one-way analysis of variance (one-way ANOVA) was employed. A significance level of (*p* < 0.05) was established for determining significant differences between datasets, and data were presented uniformly as “mean ± standard error” (mean ± SE).

## 3. Results

### 3.1. Effect of Replacing Fishmeal with Chicken Meal on the Growth Performance of Bullfrogs

The substitution of chicken meal for fishmeal did not significantly impact the final average weight or the average weight gain rate of bullfrogs (*p* > 0.5). However, replacing 50% of fishmeal with chicken meal significantly improved the total weight gain rate and survival rate while effectively reducing the feed coefficient (*p* < 0.05). Conversely, complete replacement (100%) of fishmeal with chicken meal resulted in a decrease in both survival rate and total weight gain rate, with the latter being significantly lower than that observed in the FM group (*p* < 0.05) (Table 4).

### 3.2. The Effect of Chicken Meal Replacing Fishmeal on the Amino Acid Composition of Muscle

When chicken meal was substituted for fishmeal, a reduction in the total amino acid content was observed in bullfrog muscle, including both essential (EAAs) and non-essential amino acids (NEAAs). The results indicate that the major essential amino acids present in bullfrog muscle are lysine, leucine, and arginine, among others, while the predominant non-essential amino acids include aspartic acid, glutamic acid, and alanine, among others (Table 5).

### 3.3. The Effect of Chicken Meal Replacing Fishmeal on the Antioxidant Capacity of Bullfrog Muscles

Following the partial substitution of fishmeal with chicken meal, significant effects were observed on the antioxidant enzyme activity and corresponding gene expression in bullfrogs. Specifically, when 50% of fishmeal was replaced by chicken meal, there was a significant enhancement in catalase (CAT) and total antioxidant capacity (T-AOC) in the bullfrog muscle tissue. The mRNA transcription of antioxidant genes such as *cat*, *sod*, *gst*, and *sult* was also significantly upregulated (*p* < 0.05). However, when fishmeal was completely replaced by chicken meal (100% replacement), both enzyme activities and gene expression levels exhibited a downward trend. Moreover, under complete replacement conditions, the content of malondialdehyde (MDA) in muscle tissue significantly increased (Figure 1).

### 3.4. Effect of Chicken Meal Replacing Fishmeal on the Intestinal Tissue Structure and Digestive Enzyme Activity of Bullfrogs

Following the incorporation of chicken meal as a substitute for fishmeal, significant morphological changes were observed in the intestinal tract of bullfrogs (Figure 2). Notably, the length of the intestinal villi and the thickness of the muscular layer were significantly affected by the proportion of chicken meal substitution in the feed (*p* < 0.05), with the CM50 group exhibiting the longest villi and thickest muscular layer, marking a clear difference from the other groups (*p* < 0.05) (Table 6).

Additionally, the activity of digestive enzymes in the intestine was also influenced by this substitution. Compared to the FM group, the rate of chicken meal substitution significantly altered the activities of amylase, lipase, and trypsin (*p* < 0.05). The CM50 group of bullfrogs showed significantly higher levels of amylase and lipase activities than the other groups (*p* < 0.05) (Table 6).

### 3.5. The Effect of Replacing Fishmeal with Chicken Meal on the Antioxidant Capacity and Inflammatory Indicators in the Intestine of Bullfrogs

Compared to the FM group, the different substitution ratios of chicken meal for fishmeal significantly affected the antioxidant-related indices in the intestine. Specifically, the CM50 group exhibited significantly higher activities of CAT, SOD, and T-AOC than the other groups (*p* < 0.05), while the CM100 group had a significantly higher MDA content (*p* < 0.05) (Figure 3).

Similarly, the proportion of chicken meal substituting fishmeal also significantly altered the expression of inflammatory factors and antioxidant-related genes in the intestine (*p* < 0.05). The CM50 group showed a significant increase in the expression of *il-10*, *cat*, *sod*, *gst*, and *sult* genes, accompanied by a significant decrease in *il-8* and *il-17* expression (*p* < 0.05). In contrast, the CM100 group exhibited a significant upregulation in the expression of *il-8* and *il-17*, differing significantly from the FM group (*p* < 0.05) (Figure 3).

## 4. Discussion

This study revealed that when 50% of fishmeal was substituted with chicken meal, there was a significant enhancement in the survival rate and the total weight of bullfrogs, aligning with previous research that emphasizes the benefits of moderate substitution [21,22]. Various novel proteins have shown positive effects as suitable substitutes in aquafeed, such as promoting growth, optimizing metabolism, and strengthening immunity [9,23,24], which may be attributed to the optimized balance of nutrients, complementary effects, and improvement in the physical properties of the feed. Different ingredients affect the physical characteristics of the feed, such as hardness, density, and water absorption. The optimization of the texture and palatability of mixed feeds may stimulate the appetite of bullfrogs, increase feed intake, and directly accelerate growth and weight gain [25]. Moreover, with a 50% substitution, improvements in intestinal structure and function were observed, including increased villi length, thickened intestinal walls, and enhanced digestive enzyme activity, which together provide favorable conditions for efficient nutrient absorption [26,27]. The increased length of villi expands the absorptive surface area, and the thickening of the muscular layer of the intestinal wall promotes intestinal motility. These structural optimizations may be related to the promotive effects of specific nutrients in chicken meal. Moderate substitution promotes the balance and absorbability of nutrients, enhancing digestive enzyme activity and food digestion efficiency [9]. Furthermore, an important direction for future research is to investigate whether the efficiency of substituting fishmeal with chicken meal can be further improved by supplementing specific nutrients in the feed, such as some amino acids, fatty acids, and trace elements [28,29,30,31].

The 50% chicken meal substitution enhanced antioxidant capacity, as evidenced by the increased activity of antioxidant enzymes and the expression of related genes in muscle and intestinal tissues. This suggests that chicken meal may reduce oxidative stress damage by strengthening the antioxidant defense mechanisms. Additionally, moderate substitution may activate the body’s adaptive stress response, upregulating the production and gene expression of antioxidant enzymes, enhancing the overall antioxidant status [20,32,33]. Under this substitution scheme, the downregulation of inflammatory factor gene expression in bullfrog intestinal tissues may reflect improved nutritional balance and digestive efficiency. Meanwhile, the increased activity of digestive enzymes accelerated food digestion and absorption, reducing intestinal irritation and inflammatory responses, further corroborating the positive effects of the substitution strategy [34,35,36].

Compared to the 50% partial substitution, the complete substitution of fishmeal with chicken meal significantly affected the growth performance and intestinal health indices of bullfrogs, closely related to nutritional imbalance and reduced digestive and absorptive efficiency. Fishmeal, as a core component of feed, is rich in indispensable nutrients for maintaining animal health, including amino acids with high bioavailability and specific fatty acids [37,38,39]. The complete substitution strategy may lead to a deficiency of key nutrients, particularly the insufficient supply of essential amino acids, disrupting the normal growth and physiological functions of bullfrogs [9,40]. Preliminary amino acid profile analysis confirmed a decline in amino acid content in the feed after chicken meal substitution. Additionally, bullfrogs exhibit a higher digestibility of amino acids from fishmeal compared to other feed proteins such as poultry by-product meal (chicken meal), feather meal, and soybean meal [41]. This indicates that the complete replacement of fishmeal with chicken meal may reduce the utilization efficiency of certain amino acids. The substitution effects of chicken meal in specific nutritional dimensions are not perfect; for example, this substitution increases indigestible components, burdens the intestine, induces inflammation, and results in the release of inflammatory factors [42,43]. The substantial metabolic pathway adjustments triggered by complete substitution may generate excess free radicals, exceeding the clearance threshold of the enhanced antioxidant system, ultimately inhibiting antioxidant enzyme activity and related gene transcription [44,45]. Furthermore, numerous studies have highlighted that the substitution of fishmeal can significantly impact gut microbiota, an aspect that cannot be overlooked [46,47]. The complete substitution may disrupt the microbial community homeostasis, reduce beneficial bacterial populations, lead to increased pathogen load and decreased digestive efficiency, and exacerbate intestinal inflammation [34,43,48]. Damage to the intestinal barrier function not only restricts nutrient intake but also facilitates pathogen invasion and inflammation progression, posing a compound negative impact on the health and growth of bullfrogs [49,50].

## 5. Conclusions

In summary, chicken meal as a partial substitute for fishmeal can optimize the survival rate, antioxidant capacity, and intestinal health of bullfrogs, but excessive substitution should be approached with caution to avoid negative effects due to nutritional imbalance or the absence of specific components.

## Figures and Tables

**Figure 1 animals-14-02200-f001:**
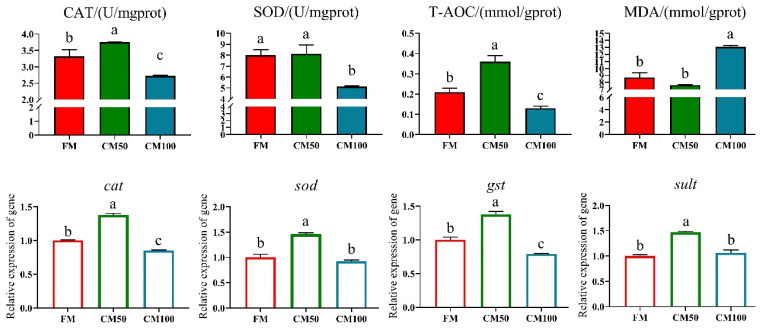
Antioxidant capabilities in muscle following the replacement of fishmeal with chicken meal in feed. Catalase (CAT), superoxide dismutase (SOD), total antioxidant capacity (T-AOC), malondialdehyde (MDA), catalase (*cat*), superoxide dismutase (*sod*), glutathione-S-transferase (*gst*), and sulfotransferase (*sult*). Genes are denoted in italicized lowercase. a–c: in the same indicator, those with the same or no letters indicate that there is no significant difference between them (*p* > 0.05).

**Figure 2 animals-14-02200-f002:**
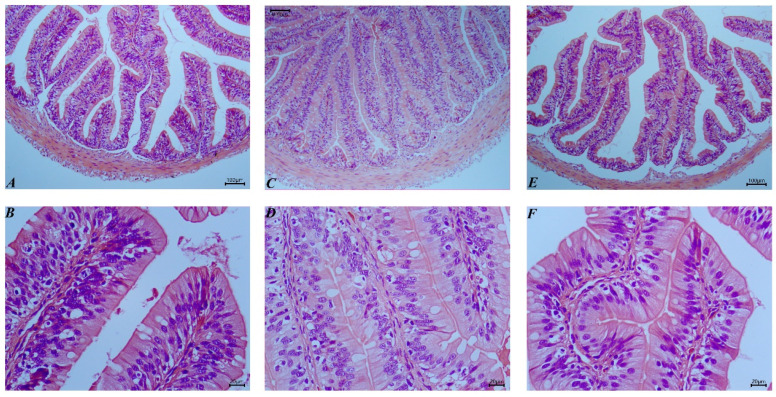
Histological observations of bullfrog intestinal tissues following the substitution of fishmeal with chicken meal (H&E staining). (**A**) FM, 100×; (**B**) FM, 400×; (**C**) CM50, 100×; (**D**) CM50, 400×; (**E**) CM100, 100×; (**F**) CM100, 400×.

**Figure 3 animals-14-02200-f003:**
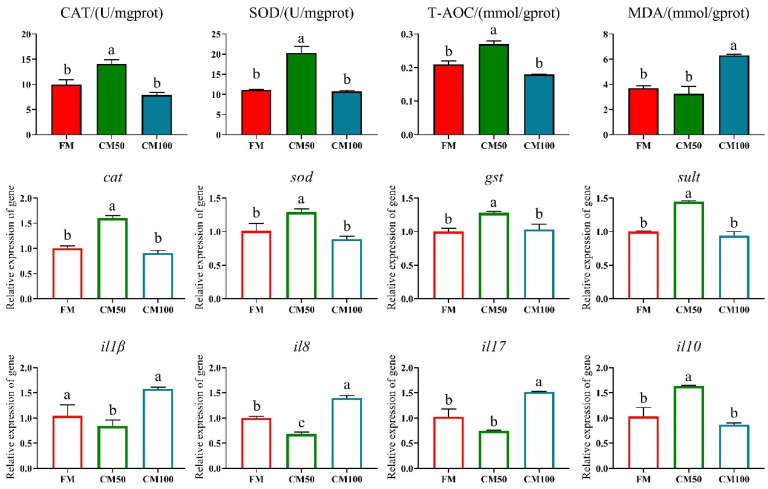
Antioxidant capacity and inflammatory markers in the intestine of bullfrogs following the substitution of fishmeal with chicken meal. The indicators include catalase (CAT), superoxide dismutase (SOD), total antioxidant capacity (T-AOC), malondialdehyde (MDA), catalase (*cat*), superoxide dismutase (*sod*), glutathione-S-transferase (*gst*), sulfotransferase (*sult*), interleukin-1β (*il1β*), interleukin-8 (*il8*), interleukin-17 (*il17*), and interleukin-10 (*il10*). Italicized lowercase denotes genes. a–c: in the same indicator, those with the same or no letters indicate that there is no significant difference between them (*p* > 0.05).

**Table 1 animals-14-02200-t001:** Composition and nutrient composition of experimental feed (%).

Ingredients	FM	CM50	CM100
Fishmeal	20.00	10.00	0.00
Chicken meal	0.00	9.80	19.80
Soybean meal	30.70	30.70	30.70
Corn protein meal	5.00	5.00	5.00
Rapeseed meal	9.00	9.00	9.00
Bentonite	2.70	2.39	1.87
Rice bran	11.00	11.00	11.00
Wheat flour	18.00	18.00	18.00
Soybean oil	2.15	1.68	1.25
Premix ^1^	0.40	0.40	0.40
Ca (H_2_PO_4_)_2_	0.50	1.09	1.67
NaCl	0.30	0.30	0.30
Choline chloride	0.25	0.25	0.25
Lysine	0.00	0.29	0.57
Methionine	0.00	0.10	0.19
Total	100.00	100.00	100.00
Moisture	7.72	7.58	7.64
Crude protein	35.80	35.78	35.78
Crude lipid	6.27	6.28	6.29
Ash	8.97	9.02	9.08

Note: ^1^ Premix consists of a blend of minerals and vitamins, providing the following per kilogram of feed: KCl 100 mg, KI (1%) 30 mg, CoCl_2_·6H_2_O (1%) 25 mg, CuSO_4_·5H_2_O 15 mg, FeSO_4_·H_2_O 200 mg, ZnSO_4_·H_2_O 200 mg, MnSO_4_·H_2_O 75 mg, Na_2_SeO_3_·5H_2_O (1%) 32.5 mg, MgSO_4_·H_2_O 1000 mg, zeolite powder 1987 mg, thiamine (VB_1_) 6 mg, riboflavin 6 mg, pyridoxine (VB_6_) 4 mg, cobalamin (VB_12_) 0.10 mg, menadione (VK_3_) 4 mg, inositol 50 mg, pantothenic acid 20 mg, niacin 25 mg, folic acid 2.5 mg, biotin 0.4 mg, vitamin A 50 mg, vitamin D 17.5 mg, vitamin E 25 mg, vitamin C 50 mg, ethoxyquin 75 mg.

**Table 2 animals-14-02200-t002:** Amino acid composition and content in feed (g/100 g).

Items	FM	CM50	CM100
Lysine	1.70	1.69	1.69
Phenylalanine	1.49	1.45	1.37
Threonine	1.16	1.13	1.07
Isoleucine	1.38	1.22	1.08
Leucine	2.65	2.55	2.44
Valine	1.58	1.41	1.25
Argnine	1.83	1.82	1.76
Methionine	0.45	0.49	0.51
Histidine	0.80	0.78	0.71
∑EAA	13.04	12.54	11.88
Aspartic acid	2.78	2.70	2.55
Glutamic acid	5.25	5.17	5.08
Cystine	0.24	0.25	0.26
Serine	1.14	1.17	1.13
Glycine	1.49	1.52	1.64
Alanine	1.73	1.71	1.64
Proline	1.86	1.84	1.91
Tyrosine	0.87	0.88	0.86
∑NEAA	15.36	15.24	15.07
Total amino acid	28.4	27.78	26.95

**Table 3 animals-14-02200-t003:** Primer sequences for gene qPCR process.

Gene	Forward Primer (5′-3′)	Reverse Primer (5′-3′)
*il-10*	GGAAGGACAGTTCAGCCCAA	CGCTGTGAAACCGAAGTAGC
*il-1β*	TCATTCGGGACAGCAGGCAGAA	GCTTCACTGGCACGGTTGTTCT
*il-8*	GCACAGCAGGCAGCAGCATT	ACAAACCACTTAACACTGGCAGGG
*il-17*	TGATAGTCACGCACTGAGTCCG	ATGTTCACCAGCCAGTCAATGC
*cat*	GATGGGAACTGGGATCTGACTGGAAA	CTGAGAGTGGATGAATGACGGGAACA
*sod*	GCATTCTATCATTGGACGCACAGCA	CCCACCAGCATTGCCAGTTATCA
*gst*	GTGTGGATTGGAAAGAAGAGGTGGTGA	TCCTAGCAAGATGGCGGAGTATGG
*sult*	GAAGACATGAAAGCGGACCTCAC	GCTCATCCTTCAGAGCTAAGCCATA
*β actin*	CATCCTTCTTGGGTATGGAATCA	TGGCATACAGGTCCTTACGGATA

Note: The genes’ corresponding CDS sequences are provided in Appendix A.

**Table 4 animals-14-02200-t004:** Growth performance of bullfrogs following the substitution of fishmeal with chicken meal in feed.

Items	FM	CM50	CM100
Initial average weight/g	44.02 ± 0.10	44.04 ± 0.20	44.03 ± 0.20
Final average weight/g	140.73 ± 0.58	135.17 ± 6.26	137.31 ± 1.04
Average weight gain rate/%	215.76 ± 1.37	206.98 ± 14.26	211.8793 ± 5.12
Total weight gain rate/%	125.41 ± 1.63 ^b^	150.66 ± 11.66 ^a^	107.31 ± 5.12 ^c^
Feed coefficient	1.19 ± 0.01 ^ab^	1.11 ± 0.06 ^b^	1.29 ± 0.03 ^a^
Survival rate/%	70.50 ± 0.65 ^b^	81.75 ± 2.78 ^a^	66.50 ± 1.94 ^b^

Letter: in the same indicator, those with the same or no letters indicate that there is no significant difference between them (*p* > 0.05).

**Table 5 animals-14-02200-t005:** Amino acid composition of bullfrog muscle following the substitution of fishmeal with chicken meal (g/100 g).

Items	FM	CM50	CM100
Lysine	1.82 ± 0.21	1.73 ± 0.12	1.67 ± 0.133
Phenylalanine	0.83 ± 0.03	0.78 ± 0.04	0.76 ± 0.05
Threonine	0.75 ± 0.04	0.68 ± 0.06	0.71 ± 0.07
Isoleucine	0.93 ± 0.05	0.87 ± 0.05	0.83 ± 0.06
Leucine	1.48 ± 0.06	1.41 ± 0.09	1.39 ± 0.13
Valine	0.92 ± 0.07	0.91 ± 0.08	0.84 ± 0.08
Argnine	1.22 ± 0.04	1.13 ± 0.07	1.17 ± 0.09
Methionine	0.29 ± 0.01	0.24 ± 0.02	0.27 ± 0.01
Histidine	0.55 ± 0.05	0.56 ± 0.04	0.53 ± 0.05
∑EAAs	8.79	8.31	8.17
Aspartic acid	1.72 ± 0.08	1.65 ± 0.10	1.63 ± 0.13
Glutamic acid	3.39 ± 0.21	3.20 ± 0.26	3.23 ± 0.25
Serine	0.70 ± 0.03	0.68 ± 0.04	0.70 ± 0.04
Glycine	0.82 ± 0.08	0.82 ± 0.05	0.89 ± 0.07
Alanine	1.00 ± 0.10	0.99 ± 0.07	0.97 ± 0.08
Proline	0.74 ± 0.07	0.70 ± 0.06	0.74 ± 0.04
Tyrosine	0.53 ± 0.03	0.47 ± 0.03	0.48 ± 0.04
∑NEAAs	8.90	8.51	8.64
Total amino acid	17.69	16.82	16.81

**Table 6 animals-14-02200-t006:** Intestinal tissue structure and digestive enzyme activity of bullfrogs after replacing fishmeal with chicken meal.

Items	FM	CM50	CM100
Villus length/μm	834.13 ± 58.05 ^a^	1374.28 ± 104.52 ^b^	773.05 ± 32.37 ^a^
Muscle layer thickness/μm	141.74 ± 0.68 ^a^	151.74 ± 0.83 ^b^	138.77 ± 2.34 ^a^
Goblet cell	29.67 ± 0.33 ^a^	32.00 ± 1.00 ^b^	28.67 ± 0.33 ^a^
Amylase (U/gprot)	3.72 ± 0.19 ^b^	4.13 ± 0.05 ^b^	2.85 ± 0.12 ^a^
Lipase (U/gprot)	2.53 ± 0.34 ^a^	5.63 ± 0.64 ^b^	2.24 ± 0.68 ^a^
Trypsin (U/gprot)	72.12 ± 0.96 ^b^	73.24 ± 0.35 ^b^	70.31 ± 0.04 ^a^

Letter: in the same indicator, those with the same or no letters indicate that there is no significant difference between them (*p* > 0.05).

## Data Availability

The data of this study are available from the corresponding authors upon reasonable request.

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
