# Peer review of "Chicken Meal as a Fishmeal Substitute: Effects on Growth, Antioxidants, and Digestive Enzymes in Lithobates catesbeianus"

_animals, 2024, doi:10.3390/ani14152200_

Round 1

Reviewer 1 Report

Comments and Suggestions for Authors

Title: ‘Intestinal Digestive Function’ This phrase needs to be modified. Only the activities of amylase, lipase, and trypsin in the intestine were determined, it was described as intestinal digestive function, which somewhat exaggerated the role of digestive enzyme activity.

L12 ‘: In’ should not be bold. Please change ‘evaluates’ to ‘was to evaluate’.

L16 ‘survival rate’ not ‘survival rates’

L17 What are the specific detrimental effects?

L20 Intestine not ‘intestines’

L36-37 Are all these aquatic products come from aquatic animals? Do they all need aquafeeds?

L39 ‘high-quality feed’ should be changed to ‘high-quality protein sources’.

L83 ‘delve into’ Please change to ‘investigate’.

L89 Please supplement the ash and moisture contents of experimental diets in Table 1.

L161 Please supplement the NCBI no. of the genes.

L178 Why are the results different for total weight gain rate average weight gain rate?

L190 Please supplement standard error of the data from amino acid composition of the muscle in the Table 4. And then, the data between treatments were subjected to the analysis of One-Way ANOVA.

L193-195 The Sentence is a little confusing. Please rewrite it.

L198 What is the function of this sult gene and is it related to antioxidant capacity?

L248 ‘revealed’ not ‘reveals’. Present the findings of the present study, generally using the past tense. Please revise it in the entire manuscript.

L254-257 The concentrations of fatty acids and micronutrients in chicken meal and fishmeal need to be provided in order to confirm your speculation. The references form 25-27 did not provide the related data, and did not the studies on chicken meal.

L260 Increase feed intake, which will directly accelerate growth and weight gain. Please change indirectly into directly.

L275-277 If the authors suggested this argue, it needed to demonstrate that the levels of selenium and vitamin E in chicken meal were higher than those in the fishmeal.

L283 What is the nutritional structure?

L293-297 Since the authors suspect that fatty acids may also be variable, why not measure them?

L299-302 Anti-nutritional factors in chicken meal? The references form 50-53 described the anti-nutritional factors in the plant proteins, such as soybean meal. In general, anti-nutritional factors are found mainly in plant proteins. The authors need to provide reliable evidence that anti-nutritional factors are present in chicken meal.

L320-322 In the conclusion, authors mentioned the ideal substitution ratio. Why was this experiment set up with just 2 levels of substitution?

Comments on the Quality of English Language

No.

Author Response

Comments and Suggestions for Authors

Comment 1. Title: ‘Intestinal Digestive Function’ This phrase needs to be modified. Only the activities of amylase, lipase, and trypsin in the intestine were determined, it was described as intestinal digestive function, which somewhat exaggerated the role of digestive enzyme activity.

Response 1. Thanks to your suggestion, we have revised the title of the manuscript.

Comment 2. L12 ‘: In’ should not be bold. Please change ‘evaluates’ to ‘was to evaluate’.

Response 2. Thank you for your suggestion, we have changed.

Comment 3. L16 ‘survival rate’ not ‘survival rates’

Response 3. Thank you for your suggestion, we have changed.

Comment 4. L17 What are the specific detrimental effects?

Response 4. Thank you for your suggestion, the expression here was not clear and we have modified it, it refers to the CM100 group having an opposite effect to the CM50 group (which tends to be bad).

Comment 5. L20 ‘Intestine’ not ‘intestines’

 Response 5. Thank you for your suggestion, we have changed.

Comment 6. L36-37 Are all these aquatic products come from aquatic animals? Do they all need aquafeeds?

 Response 6. Thank you for pointing out the problem, not all aquatic products require aquafeeds, we have made changes to more specifically indicate that aquafeeds are required for captive bred fish.

Comment 7. L39 ‘high-quality feed’ should be changed to ‘high-quality protein sources’.

 Response 7. Thank you for your suggestion, we have changed.

Comment 8. L83 ‘delve into’ Please change to ‘investigate’.

 Response 8. Thank you for your suggestion, we have changed.

Comment 9. L89 Please supplement the ash and moisture contents of experimental diets in Table 1.

 Response 9. Thanks to your suggestion, we have added ash and moisture contents to Table 1.

Comment 10. L161 Please supplement the NCBI no. of the genes.

Response 10. Thanks to your suggestion, we have provided a supplementary file (Supplementary file 1) containing detailed cds sequences corresponding to genes for comparison in the NCBI database.

Comment 11. L178 Why are the results different for total weight gain rate average weight gain rate?

 Response 11. I apologize for the misunderstanding of our description. total weight gain rate and average weight gain rate are calculated differently (this is included in the manuscript). total weight gain rate is calculated using the total weight of the bullfrogs in the pre- and post-growth periods, so survival rate has a significant impact on this metric. Bullfrogs in the CM50 group had significantly higher survival rates, so the total weight gain rate would be higher, even though the difference in average weight was not significant.

Comment 12. L190 Please supplement standard error of the data from amino acid composition of the muscle in the Table 4. And then, the data between treatments were subjected to the analysis of One-Way ANOVA.

  Response 12. Thanks to your suggestion, we have added the relevant data.

Comment 13. L193-195 The Sentence is a little confusing. Please rewrite it.

 Response 13. Thank you for your suggestion, we have changed.

Comment 14. L198 What is the function of this sult gene and is it related to antioxidant capacity?

Comment 14. Thank you for asking this question.The Sulfotransferase, SULT gene plays an important role in the detoxification of metabolic environmental toxins.The effect of SULT on antioxidant capacity is mainly seen in its metabolism and regulation of antioxidants in the body.SULT enzymes are capable of affecting the stability and activity of a number of molecules with antioxidant effects. SULT enzymes can affect the stability and activity of some molecules with antioxidant effects, such as the sulfation of some endogenous antioxidants that may alter their activity. For example, the sulfation of some endogenous antioxidants may alter their activity, and the metabolism of some toxins may indirectly affect their oxidative stress on the body.

Comment 15. L248 ‘revealed’ not ‘reveals’. Present the findings of the present study, generally using the past tense. Please revise it in the entire manuscript.

  Response 15. Thank you for your suggestion, we have changed.

Comment 16. L254-257 The concentrations of fatty acids and micronutrients in chicken meal and fishmeal need to be provided in order to confirm your speculation. The references form 25-27 did not provide the related data, and did not the studies on chicken meal.

 Response 16. Thank you for your suggestion, we did not test that data directly and therefore could not come to that conclusion, so we are revising that content.

Comment 17. L260 Increase feed intake, which will directly accelerate growth and weight gain. Please change indirectly into directly.

  Response 17. Thank you for your suggestion, we have changed.

Comment 18. L275-277 If the authors suggested this argue, it needed to demonstrate that the levels of selenium and vitamin E in chicken meal were higher than those in the fishmeal.

 Response 18. Thank you for your suggestion, we did not test that data directly and therefore could not come to that conclusion, so we are revising that content.

Comment 19. L283 What is the nutritional structure?

Response 19. Thank you for your comment, the content was not expressed clearly enough and we have made changes.

Comment 20. L293-297 Since the authors suspect that fatty acids may also be variable, why not measure them?

 Response 20,Thank you for your suggestion, we did not test that data directly and therefore could not come to that conclusion, so we are revising that content.

Comment 21. L299-302 Anti-nutritional factors in chicken meal? The references form 50-53 described the anti-nutritional factors in the plant proteins, such as soybean meal. In general, anti-nutritional factors are found mainly in plant proteins. The authors need to provide reliable evidence that anti-nutritional factors are present in chicken meal.

 Response 21 Thank you for your suggestion, our expression is not accurate enough, antinutritional factors are mainly found in plant proteins, they affect the absorption and utilization of nutrients in proteins by animals, whereas in animal proteins there are usually very few antinutritional factors, and more often than not it is the residuals of some hormones or antibiotics and other medicines, so we modify the content.

Comment 22. L320-322 In the conclusion, authors mentioned the ideal substitution ratio. Why was this experiment set up with just 2 levels of substitution?

Response 22 Thank you for your question, the original intention of the experimental design was not to come up with the most suitable substitution rate through one experiment, because it would require designing more substitution ratios thus leading to an increase in various costs and inputs. Instead, our aim was more to first verify that chicken meal can effectively replace fish meal in bullfrog feed, and after determining this result, then try to further verify the suitable replacement ratio in the following experiments. We modified this expression.

Reviewer 2 Report

Comments and Suggestions for Authors

Title of the manuscript is too long. Author are advised to make it shorter and more meaningful. Authors are suggested to mention the scientific name of the species. 

Why the fish meal needs to be incorporated in bullfrog diet?

Line 33-40 cannot be the part of this article as it is discussing about fish/shrimp culture or general aquaculture. Author needs to be specific in discussing only related things related to bullfrog culture. 

The article does not give proper reason that why it is required that fish meal should be replaced by chicken meal in the diet of bullfrog. How can it reduce the feed cost as giving chicken meal is a costly affair as compared to fishmeal. 

Authors are suggested to include a paragraph about chicken meal and how it was prepared without which this article does not have any value.

Looking into the formulation raises serious concern such as, why benonite quantity was reduced into the diet. Why lysine and methionine was added in CM50 and CM 100 diets.

Authors are suggested to give complete nutritional profile of chicken meal including amino acid, fatty acid and anti-nutrients if any.

What was the basis for deciding their ration at 3%?

Authors are suggested to give complete methodology for line 142-145.

What is reason for poor absorption of Lysine and methionine in CM50 and CM100 even after additional addition of lysine and methionine in these diets. 

Growth performance of bullfrogs must be shown in data form. Authors are suggested to change fig. 1 in to tabular for better clarity.

I see that there is no significant difference in final average weight and average weight gain rate than how come total weight gain rate varied significantly. 

I see that survival rate of fishmeal and CM100 is significantly lower than why author is targeting to replace fish meal in bullfrog diet.

Similarly, fishmeal-based diet has significantly lower antioxidant enzyme profile than why come author is suggesting replacing fish meal. 

The conclusion section is totally misleading. Authors are suggested to do a thorough interpretation of data and rewrite the entire result and conclusion section. 

Comments on the Quality of English Language

The author is requested to do extensive English editing from a native English speaker. 

Author Response

Comments and Suggestions for Authors

Comment 1. Title of the manuscript is too long. Author are advised to make it shorter and more meaningful. Authors are suggested to mention the scientific name of the species. 

Response 1. Thanks to your suggestion, we have revised the title of the manuscript.

Comment 2. Why the fish meal needs to be incorporated in bullfrog diet?

Response 2. Bullfrogs, as carnivorous animals, feed mainly on insects and other animals in nature. In China, bullfrogs are managed as aquatic animals, and Chinese aquatic feeds (carnivorous) tend to contain a relatively high amount of animal protein to meet the growth needs of the animal. Fishmeal is recognized as a high quality feed protein in China's aquafeed industry, so feed companies have added some fishmeal to bullfrog feeds, while indeed achieving better growth results, but this also comes with higher feeding costs.

Comment 3. Line 33-40 cannot be the part of this article as it is discussing about fish/shrimp culture or general aquaculture. Author needs to be specific in discussing only related things related to bullfrog culture. 

Response 3. Thank you for raising this issue. The content of this paragraph is mainly to illustrate the importance of aquaculture and the importance and urgency of the issue of fishmeal substitution in aquafeeds (the same exists in bullfrog aquaculture as bullfrogs are managed as aquatic products in China). We are aware of this problem and have made some deletions, as well as merging the first and second paragraphs.

Comment 4. The article does not give proper reason that why it is required that fish meal should be replaced by chicken meal in the diet of bullfrog. How can it reduce the feed cost as giving chicken meal is a costly affair as compared to fishmeal. 

Response 4. Thank you for asking this question, the fact is that fishmeal is more expensive than chicken meal for the protein ingredients used in aquafeeds. As of June 2024, the import price of chicken meal in China is approximately $1,130 per ton, compared to around $2,200 per ton for Peruvian super fishmeal, thereby making chicken meal a cost-effective alternative.

Comment 5. Authors are suggested to include a paragraph about chicken meal and how it was prepared without which this article does not have any value.

Response 5. Thanks to your suggestion, we have added the paragraph

Comment 6. Looking into the formulation raises serious concern such as, why benonite quantity was reduced into the diet. Why lysine and methionine was added in CM50 and CM 100 diets.

Response 6. Thank you for bringing up this question. The adjustment of bentonite in the feed is targeted at ensuring that the total volume of the feed stays at 100% subsequent to alterations in other components. Given that feed formulations endeavor to uphold equal proportions of crude protein and crude lipid across different batches, and bearing in mind that fishmeal and poultry by-product meal do not possess identical concentrations of these nutrients, a straight weight-for-weight substitution would not be sufficient during protein-equivalent replacement. Thus, bentonite is utilized to counterbalance the total feed volume.

Similarly, soybean oil is applied to guarantee equivalent quantities of crude lipid. Upon substituting fishmeal with poultry by-product meal, it was observed by New Hope Group (although not publicly disclosed) that the latter exhibits lower concentrations of lysine and methionine relative to fishmeal. To reconcile this disparity, they prudently augmented the feed with suitable quantities of lysine and methionine to maintain as close a parity as possible in these amino acids. The underlying intent was to augment the effectiveness of poultry by-product meal as a substitute for fishmeal without instigating a pronounced escalation in price, considering that the expense of amino acid supplementation is inferior to that incurred by fishmeal.

This methodology is commonplace in commercial feed production procedures, and the formulation employed in this study is an adaptation derived from such commercial practices. The rationale for specifically supplementing lysine and methionine is that these two amino acids are universally acknowledged within the industry as the most crucial and most extensively researched essential amino acids for the majority of aquatic organisms.

Comment 7. Authors are suggested to give complete nutritional profile of chicken meal including amino acid, fatty acid and anti-nutrients if any.

Response 7. We supplemented the crude protein and crude fat content of chicken meal in 2.1. experimental feed. But for the other nutritional profiles we don't have more precise test values.

Comment 8. What was the basis for deciding their ration at 3%?

Response 8. Thank you for raising this point. In the original draft, the figure of 3% was indeed imprecise; it should have been specified as a range of 3% to 5%, which we have now corrected. This range represents a standard reference for feeding rates, although in practice, our actual feeding quantities are adjusted based on water temperature and the frogs' appetite. Typically, we feed until the bullfrogs appear visibly sated, and any uneaten feed remaining after a 10-minute period is removed to prevent waste and maintain water quality.

Comment 9. Authors are suggested to give complete methodology for line 142-145.

Response 9. Thank you for your suggestion, we have added this content.

Comment 10. What is reason for poor absorption of Lysine and methionine in CM50 and CM100 even after additional addition of lysine and methionine in these diets. 

Response 10. Thank you for bringing up this issue. Despite supplementing lysine and methionine in the CM50 and CM100 groups, the absorption of these two amino acids still appears to be suboptimal in these groups. This is due to the fact that the digestibility of these amino acids from poultry by-product meal is lower in bullfrogs than when these amino acids are sourced from fishmeal. We have addressed this point in the discussion section of our manuscript and cited relevant literature to support this explanation.

Comment 11. Growth performance of bullfrogs must be shown in data form. Authors are suggested to change fig. 1 in to tabular for better clarity.

Response 11. Thank you for your suggestion, we have made the necessary modifications.

Comment 12. I see that there is no significant difference in final average weight and average weight gain rate than how come total weight gain rate varied significantly. 

Response 12. We apologize if our description led to some confusion. The calculation methods for the total weight gain rate and the average weight gain rate are indeed different, a point that is already included in our manuscript. The total weight gain rate is calculated using the total weight of the bullfrogs at the beginning and end of the experiment. Consequently, survival rate has a significant impact on this metric. Since the survival rate of the CM50 group was notably higher than in the other groups, its total weight gain rate was consequently higher, even if there wasn't a substantial difference in the average weight gain.

Comment 13. I see that survival rate of fishmeal and CM100 is significantly lower than why author is targeting to replace fish meal in bullfrog diet.

Response 13. We apologize if our previous description caused any misunderstanding; we did not provide specific details regarding the prices of fishmeal and poultry by-product meal in our earlier manuscript. In reality, the price of fishmeal used in aquaculture feeds in China is significantly higher than that of poultry by-product meal. The objective of this study was to investigate the feasibility of substituting poultry by-product meal for fishmeal in order to reduce feed costs. Prior to our research, we were unaware that replacing 50% of the fishmeal with poultry by-product meal could markedly improve the survival rate of bullfrogs.

Comment 14. Similarly, fishmeal-based diet has significantly lower antioxidant enzyme profile than why come author is suggesting replacing fish meal. 

Response 14. Thank you for your question, as in “Response 13.”, the purpose of using chicken meal instead of fishmeal is to reduce the cost, and we can't know the real result after substitution until we experiment.

Comment 15. The conclusion section is totally misleading. Authors are suggested to do a thorough interpretation of data and rewrite the entire result and conclusion section. 

Response 15. Thank you for asking this question, the content of the previous conclusion is indeed not appropriate, because chicken meal significantly increased the survival rate as well as the total weight gain rate of bullfrogs after replacing 50% of fishmeal, but the effect on the mean weight gain rate was not significant. So we modified the conclusion to describe only the significant effect on survival rate.

Comment 16. Comments on the Quality of English Language

The author is requested to do extensive English editing from a native English speaker. 

Response 16. Thank you very much for your suggestions, we have invited teachers who specialize in English to correct the language problems in our manuscript.

Round 2

Reviewer 2 Report

Comments and Suggestions for Authors

The manuscript is nicely revised.